# First Molecular Evidence of Equine Herpesvirus Type 1 (EHV-1) in Ocular Swabs of Clinically Affected Horses

**DOI:** 10.3390/v17060862

**Published:** 2025-06-18

**Authors:** Beatriz Musoles-Cuenca, Miguel Padilla-Blanco, Valentina Vitale, Teresa Lorenzo-Bermejo, María de la Cuesta-Torrado, Beatriz Ballester, Elisa Maiques, Consuelo Rubio-Guerri, Ana Velloso Alvarez

**Affiliations:** 1Department of Biomedical Sciences, Faculty of Health Sciences, Universidad Cardenal Herrera-CEU, CEU Universities, 46115 Valencia, Spain; beatriz.musolescuenca@uchceu.es (B.M.-C.); teresa.lorenzobermejo@uchceu.es (T.L.-B.); beatriz.ballesterllobell@uchceu.es (B.B.); emaiques@uchceu.es (E.M.); 2Department of Pharmacy, Faculty of Health Sciences, Universidad Cardenal Herrera-CEU, CEU Universities, 46115 Valencia, Spain; miguelpadillablanco54@gmail.com; 3Viral Immunology Laboratory, Molecular Biomedicine Department, Margarita Salas Center for Biological Research (CIB-CSIC), 28040 Madrid, Spain; 4Department of Animal Medicine and Surgery, University Cardenal Herrera-CEU, CEU Universities, 46115 Valencia, Spain; valentina.vitale@uchceu.es (V.V.); maria.de2@uchceu.es (M.d.l.C.-T.)

**Keywords:** EHV-1, EHM, horses, Valencian Community 2023 outbreak, ocular swabs

## Abstract

Equine Herpesvirus Type 1 (EHV-1) is a significant pathogen within the *Alphaherpesvirinae* subfamily, causing respiratory disease, abortions, and, in severe cases, equine herpesvirus myeloencephalopathy (EHM). While nasal swabs and blood samples are commonly used for real-time polymerase chain reaction (RT-PCR) diagnosis, variability in viral shedding necessitates exploring additional sample types. This study reports the first molecular detection of EHV-1 in ocular swabs from naturally infected horses during an outbreak in the Valencian Community in 2023. Nasal and ocular swabs were collected from ten symptomatic horses and analyzed via RT-PCR. EHV-1 was detected in all cases, with higher viral loads in nasal samples. Although nasal swabs remain the most reliable sample for EHV-1 detection, the presence of viral DNA in tear fluid suggests a previously unrecognized route of viral shedding. These findings support further investigation into the role of ocular secretions in the pathogenesis and epidemiology of EHV-1. Additional studies are needed to determine the clinical relevance and potential utility of ocular swabs in specific outbreak scenarios.

## 1. Introduction

Equine Herpesvirus Type 1 (EHV-1), also referred to as equine rhinopneumonitis virus, is a significant pathogen in the equine industry. It has recently caused devastating outbreaks worldwide, associated with adverse effects in the equine industry, and substantial economic losses [1,2]. Classified within the *Alphaherpesviridae* subfamily, specifically in the genus *Varicellovirus* [3], this virus was initially identified in 1966 in cases involving abortion and paralysis. EHV-1 is recognized as one of the nine identified equine herpesviruses to date, with its discovery contributing to a broader understanding of herpesvirus diversity in equines [3,4,5].

The widespread prevalence of EHV-1 infection is attributed to its high incidence along the different stages of equine life, being able to induce latent infections and reactivations on 80% of the equine population under stressful conditions [6,7,8,9]. EHV-1 acts as an omnipresent pathogen, causing severe effects ranging from respiratory conditions to abortions, neonatal foal mortality, and neurological disorders, resulting in equine herpesvirus myeloencephalopathy (EHM) [7,8,9]. The primary onset of EHV-1 infection occurs through the respiratory route, involving replication within the respiratory mucosa. This leads to subsequent dissemination to adjacent tissues, resulting in cell-associated viremia followed by vasculitis and thrombi formation, which can lead to potential neurological complications or abortions [10]. In addition, EHV-1 has also reported to provoke severe ocular lesions (typically chorioretinitis, keratoconjunctivitis, or uveitis) that can induce extensive retinal destruction and blindness [11,12,13,14]. Remarkably, a recent study demonstrated that 50–90% of horses experimentally infected with EHV-1 developed ocular lesions, highlighting the importance of this disease in the ocular vasculature [4].

Monitoring viral shedding is crucial to allow safe transportation or movement of uninfected horses, whereas positive horses are quarantined until they become negative [15,16]. Precise diagnosis, particularly during outbreaks, is achieved through real-time PCR (RT-PCR), emphasizing the significance of timely and optimal sample collection to confirm viral presence [15,17,18]. In the detection of EHV-1 in horses, nasopharyngeal swabs are widely considered the ideal sample for detecting the virus in nasal secretions, offering precise and durable detection [19,20]. However, emerging as a viable and less invasive alternative, particularly in field settings, nasal swabs have shown comparable effectiveness in virus detection and, at times, increased sensitivity to low viral loads [21,22]. Horses displaying overt signs usually present a high concentration of detectable EHV-1 DNA in nasal secretions as well as in blood via RT-PCR [18,23,24,25]. Conversely, subclinical infected horses mainly exhibit the presence of EHV-1 just in nasal secretions [26]. Additionally, recent studies have shown that urine can be used to detect EHV-1 DNA in naturally infected horses with neurological signs of EHM [16]. Given the high prevalence of the virus and its consequences, understanding the shedding of virus elimination in both symptomatic and subclinical animals is vital for preventing transmission and identifying new methods for virus detection.

As previously mentioned, EHV-1 has the potential to affect the ocular system by targeting its vascular network, leading to chorioretinopathies. Experimental studies and clinical observations have provided valuable insights into the possibility of EHV-1 excretion in tears fluid, emphasizing the importance of investigating this potential transmission route to improve the management and control of infections in equine populations [7,12,22]. In this context, the use of ocular samples as a diagnostic tool is particularly relevant, especially in cases where nasal swabs cause discomfort to horses or their owners [27]. This study aims to investigate the detection of EHV-1 in ocular tears fluid and compare the viral load found in these samples to that in conventional nasal swabs. To achieve this, comprehensive clinical and experimental data were collected from horses admitted to the Veterinary Clinical Hospital UCH-CEU (HCV-CEU) during the EHM outbreak that occurred in February 2023 in the Valencian Community, Spain. To the best of our knowledge, this is the first study reporting the presence of EHV-1 in ocular tears fluid from naturally infected horses. Remarkably, no other equine herpesvirus has been reported to be present in eye discharges from any equid species, highlighting the key message of the current study.

## 2. Materials and Methods

### 2.1. Study Design

In this study, data were gathered from horses hospitalized at University Veterinary Hospital (HCV-CEU) after the EHV-1 outbreak during the international show jumping events sanctioned by the International Equestrian Federation (FEI).

During the evaluation and treatment of horses at HCV-CEU, supervised by the veterinary team, clinical disease variables as well as laboratory-analyzed samples were examined to identify key patterns and determinants. The admission criteria for horses suspected of EHV-1 from the equestrian show jumping event were based on disease severity, assessed using clinical criteria agreed upon by veterinarians present at the competition and the equine clinical service of HCV-CEU. Horses eligible for admission to HCV-CEU included those exhibiting ataxia with a grade ≥ 3/5 according to the modified Lahunta and Mayhew scale [28], altered mental status, and/or cranial nerve dysfunction. Additional admission criteria encompassed persistent fever; difficulty urinating, requiring permanent urinary catheterization; signs of colic; or confirmed contact with infected animals.

Upon admission, a comprehensive medical history was obtained for each patient by compiling information provided by competition veterinarians and horse owners. This information included breed, sex, age, vaccination status upon arrival at the show jumping event, type of clinical signs associated with the disease, previous treatments received, days of viremia, and days since the first neurological signs were identified. For consistency, Day 1 of the disease was defined as the first day after the onset of disease.

This approach allowed for rigorous, evidence-based selection of admitted horses, ensuring that the most severe and clinically relevant cases received detailed monitoring and thorough analysis at HCV-CEU.

### 2.2. Description of Critical Methods

#### 2.2.1. Sample Collection

Samples were obtained from multiple sources throughout the course of monitoring the EHV-1 outbreak. For sample collection, positive collaboration of the horse was always required. Daily ocular and nasal swabs were obtained and placed in sterile 10 mL tubes without any medium using a 15 cm long polyester-tipped swab. Nasal swabs were collected from the nasal cavity as an alternative to nasopharyngeal swabs [22]. The swab was inserted completely into the ventral meatus, contacting the mucosal surface for at least 5 s. In the case of ocular swabs, the upper eyelid was carefully held open, and the swab was gently inserted into the lower conjunctival sac, ensuring it was moistened with tear fluid. Both nasal and ocular samples were taken, respectively, from both nostrils and eyes using a separate swab for each sample type to avoid any potential cross-contamination.

Sampling occurred from 21 February to 10 March 2023, with daily samples collected per horse throughout their hospitalization period. This approach ensured continuous monitoring, whether observed at the equestrian venue or upon admission to the hospital. Nasal and ocular swabs were stored at 4 °C and transported to the laboratory of the CEU Cardenal Herrera University (Valencia), where they were analyzed within 24–48 h.

In addition to sample collection, detailed clinical information was recorded, including gender, age, breed, fever onset, neurological signs, and associated complications on EHV-1 patients like vascular and urinary complications [14]. These data were gathered and recorded, together with the timing of sample collection.

#### 2.2.2. DNA Isolation and Molecular Analysis

Both swab samples were immersed in 600 μL of phosphate-buffered saline buffer (PBS), and total DNA extraction was carried out using the NZY Tissue gDNA Isolation kit (NZYtech, Lisbon, Portugal), specifically designed for swab samples, following the manufacturer’s instructions. The eluted DNA was suspended in 50 μL of molecular biology water and stored at −20 °C until real-time PCR (RT-PCR) was conducted to detect and perform a relative quantification of the viral load by determining the Cycle Thresholds (Ct). The RT-PCR assay focused on 106 bp conserved type-specific segments within the EHV-1 gene for glycoprotein B (gB). The forward primer, from nucleotide positions 1209 to 1230, has the sequence 5′-CAT ACG TCC CTG TCC GAC AGA T-3′. The reverse primer binds to positions 1314 to 1295 of the same gene and has the sequence 5′-GGT ACT CGG CCT TTT GAC GAA-3′. A 25-nucleotide probe complementary to the segment spanning positions 1275 to 1250 of the *gB* gene was used, with the sequence 5′-TGA GAC CGA AGA TCT CCT CCA CCG A-3′. This probe was labeled at the 5′ end with the reporter dye 6-carboxyfluorescein (FAM) and at the 3′ end with the Black Hole Quencher dye [17]. The RT-PCR mixture comprised 20 μL of NZYSupreme qPCR Probe Master Mix (2×) (NZYtech, Portugal), 10 pmol of each primer, 2.5 pmol of the probe, and 50 ng of total DNA, resulting in a final volume of 40 μL. The protocol consisted of (i) 5 min at 95 °C and (ii) 40 cycles of a sequence of 5 s at 95 °C and 45 s at 60 °C. Amplification and fluorescence detection were performed on a QuantStudio™ 5 Real-Time PCR System (Applied Biosystems, Thermo Fisher Scientific, Foster City, CA, USA). The primers and probe used were approved by the UK World Organization for Animal Health (WOAH) Reference Laboratory in accordance with the ISO 17025 quality system. As a positive control, samples from the previous EHV-1 outbreak in Spain (February 2021) were used, while molecular biology water served as a negative control [29].

#### 2.2.3. Statistical Analysis

For the statistical analyses, GraphPad Prism 10 was employed. In all cases, *p*-value was calculated from a 2-sided test using 0.05 as the significance level.

## 3. Results

As consequence of the 2023 outbreak, a total of 10 horses were hospitalized. Breeds from the horses affected included Anglo-European (n = 1), Zangersheide (n = 1), Dutch Warmblood (n = 1), Selle Français (n = 4), Belgian Warmblood (n = 1), Holsteiner (n = 1), and Irish Sport Horse (n = 1). Of these, there were two stallions, four geldings, and four mares, with an average age of 9.2 years (range: 6–12 years). The horses originated from various countries: Belgium (n = 1), France (n = 4), United Kingdom (n = 1), Norway (n = 1); the Netherlands (n = 1), and Ireland (n = 2).

A total of 69 nasal swabs and 63 ocular swabs were collected from 25 February to 10 March 2023, and analyzed using RT-PCR (Table 1). During the hospitalization period, all 10 horses tested positive for EHV-1 viral DNA in at least one nasal swab, resulting in a 100% detection rate at the individual level. However, when considering the total number of nasal swabs collected, the positivity rate was 81.16% (56/69; Table 2), indicating that not every nasal swab collected from each horse tested positive during the whole period.

In contrast, 8 out of 10 horses (80%; all horses except for numbers 1 and 2) tested positive for viral DNA in at least one ocular swab, demonstrating a high detection rate at the individual level. However, the overall positivity rate for the total ocular swabs collected was 53.97% (34/63; Table 2), reflecting a lower consistency of viral DNA detection in ocular samples compared to nasal swabs. The chi-square test revealed that more infection cases were detected using the nasal swab technique compared to the ocular swab (*p*-value < 0.001).

These findings highlight that while ocular swabs can identify EHV-1 in a significant number of cases, their detection efficiency is lower than that of nasal swabs when considering the total number of samples analyzed. Among the 60 samples for which both nasal and ocular swabs were collected, 30 tested positive in both swab types. Additionally, 17 samples were positive only in the nasal swab and negative in the ocular swab, while just 2 samples were positive only in the ocular swab and negative in the nasal swab (Table 1).

The viral material collected from positive samples of both swab types was evaluated using the RT-PCR test (see Materials and Methods), with results expressed in Ct values. A total of 56 nasal swabs tested positive for EHV-1, while this number decreased to 34 for the ocular swabs. The mean Ct values were 32.63 ± 0.59 and 34.65 ± 0.64 for nasal and ocular swabs, respectively (data expressed in mean ± standard error; Table 2). Student’s *t*-test indicated that the difference in suggested viral loads between the two types of swabs was statistically significant (*p*-value = 0.029) (Figure 1A). Remarkably, both variables showed a moderate positive correlation (Pearson correlation coefficient = 0.414; *p*-value = 0.023; Figure 1B), indicating that the more viral load increases in nasal swabs, the more it also increases in ocular swabs.

While initial findings showed that nasal swabs had higher positivity rates and lower Ct values than ocular swabs, further analysis was conducted to examine similarities between these sampling methods. The study aimed to determine whether EHV-1 was shed via the ocular route and to better understand the virus’s shedding dynamics and epidemiology by comparing ocular samples with the reference nasal method. For this purpose, the 8-day surveillance period was divided into their two halves, hereinafter referred to as early and advanced infection stages (from day 1 to day 4 and from day 5 to day 8, respectively; Table 2). Analysis of RT-PCR results, expressed as Ct values, revealed statistically significant differences between nasal and ocular swabs during the early infection period (*p*-value = 0.032). Particularly, early-stage nasal swabs exhibited suggested higher viral loads (mean Ct value = 32.31 ± 0.85) compared to ocular swabs (mean Ct value = 35.03 ± 0.84). However, no statistically significant difference was observed between the two swab types during the advanced-stage (*p*-value= 0.779) (Figure 1C).

## 4. Discussion

This study reports the detection of EHV-1 DNA in ocular tears fluid from naturally infected horses, suggesting a previously unknown viral shedding. This novel finding highlights the potential significance of ocular secretions in EHV-1 transmission and practices. In this study, despite detecting EHV-1 DNA in tears fluid samples, none of the infected horses exhibited clinical ophthalmic signs.

This is consistent with earlier experimental studies, which showed that horses which developed chorioretinal lesions often do so without corneal disease or uveitis, and these lesions are only observed via specialized imaging techniques such as ocular fundus photography and fluorescein angiography [12]. Unfortunately, these advanced ophthalmic techniques were not applied in the present study, and thus the presence of such subclinical lesions cannot be ruled out.

Interestingly, most ocular infections are subclinical and rarely lead to loss of function or even immediate signs, as ocular lesions associated with EHV-1 do not appear until 3 weeks–3 months post-infection [12]. Therefore, the detection of EHV-1 in ocular tears since the initial stages of the infection provides a unique opportunity to study the virus interaction with the vascular endothelium and serves as a surrogate model to study the pathogenesis of EHM, contributing valuable insights into this serious condition [18]. Ocular shedding of alphaherpesviruses is well-documented in several animal species. In bovine herpesvirus type 1 (BoHV-1), ocular shedding occurs during both primary infection and reactivation, often associated with conjunctivitis and keratoconjunctivitis [30]. Feline herpesvirus type 1 (FeHV-1) is a common cause of feline upper respiratory disease and conjunctivitis, with ocular shedding frequently observed, especially under stress-induced reactivation [31]. Canine herpesvirus (CHV-1) can also be shed in ocular secretions, particularly in puppies or immunocompromised dogs, although it is less common compared to other routes like nasal shedding [32]. Collectively, these findings highlight tears as a valuable biological sample for the detection of several alphaherpesviruses.

Surveillance of EHV-1 has been conducted using a variety of sample types, including cerebrospinal fluid [33], blood and semen [34], rectal swabs or feces [27,35], urine [16], muzzle/nasal cavities [27], and environmental swabs from feeders and water troughs [27]. However, none of these have demonstrated the same diagnostic sensitivity as nasopharyngeal or nasal swabs, which remain the gold standard for EHV-1 detection [10,18,27,34,35]. In our study, the positivity rate for EHV-1 in nasal swabs was approximately 81%, whereas it dropped to 54% in ocular swabs. Although this latter rate is significantly lower than that obtained using the gold standard detection procedure, it is still considerably higher than those reported for other noninvasive sampling methods. For instance, less than 20% of the rectal and urine samples collected in two separate studies resulted positive for EHV-1, whereas nasal swabs yielded positivity rates exceeding 70% [16,27]. In another study comparing EHV-1 detection in blood and semen samples from stallions, only 13% of semen samples were positive, while blood samples (a more invasive method) showed a positivity rate of 62% [34]. Therefore, although our findings align with these studies, as nasal swabs consistently identified more positive cases, tear fluid demonstrated a higher positivity rate than other noninvasive sample types, supporting its potential as a practical and less invasive option for EHV-1 surveillance in horses. Furthermore, during the advanced stage of infection, the viral load detected in ocular swabs was comparable to that of nasal samples. While RT-PCR detects viral DNA but not necessarily the viable virus, the consistent detection of EHV-1 in tear fluid—reported here for the first time—suggests a previously unrecognized shedding route and provides a basis for further investigation. Due to the relatively high Ct values (>30), additional analyses such as viral isolation or strain typing were unfeasible [1,18].

Although ocular swabs cannot currently replace nasal swabs as a primary diagnostic tool, their minimally invasive nature may offer practical advantages in stress-sensitive contexts, such as outbreak investigations or when nasal sampling is impractical. However, further studies with larger cohorts and inclusion of viral isolation or strain typing are required to validate the sensitivity, reliability, and clinical relevance of tear fluid sampling across different infection stages.

## 5. Conclusions

This study provides the first evidence of EHV-1 DNA detection in ocular tear fluid from naturally infected horses. Although nasal swabs remained the most effective sample type, yielding higher positivity rates and viral load, ocular swabs showed comparable Ct values during the advanced stage of infection. These findings suggest that while nasal swabs should continue to be considered the gold standard, ocular swabs could serve as a complementary sampling option under certain conditions, particularly when nasal sampling is not feasible or when ocular lesions are present. Further validation in larger outbreak-style cohorts is necessary before recommending tear fluid sampling for field use.

## Figures and Tables

**Figure 1 viruses-17-00862-f001:**
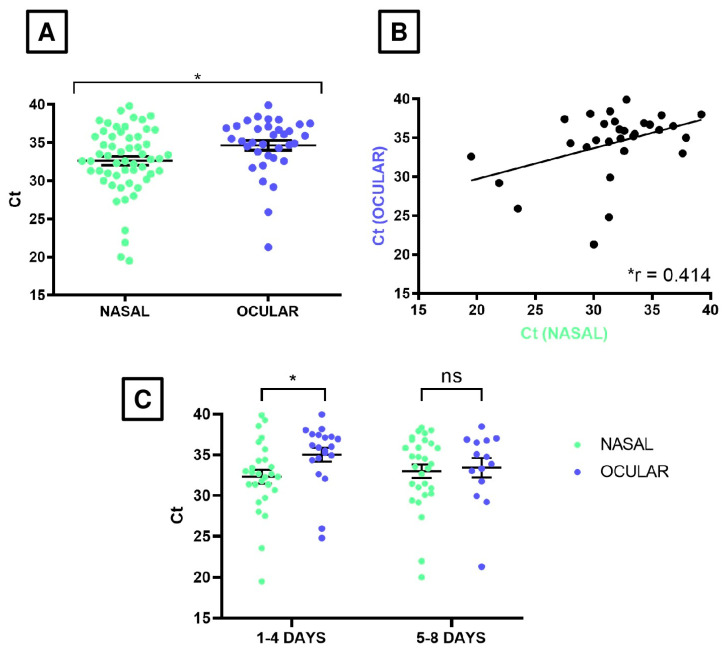
Relative quantification of viral loads by measuring the Ct values of nasal and ocular swab samples obtained via RT-PCR. (**A**) Comparison between both swabs for all the available samples. (**B**) Linear correlation (defined by the Pearson correlation coefficient, referred as “r”; r = 0.414, *p*-value = 0.029) between both types of samples when paired data (nasal and ocular swab samples collected in the same infection day) were available. (**C**) Comparison of Ct values between the nasal and ocular swab samples categorized by the two infection stages: early and advanced stages of infection (1–4 and 5–8 days of the disease). In (**A**,**C**), horizontal and error lines denote mean ± standard error. A Ct value of 40 was the cutoff for positivity, as this was the maximum number of amplification cycles used in the RT-PCR protocol. ns indicates a non-significant difference, whereas the asterisk in (**A**,**C**) shows a statistically significant difference (*p*-value = 0.029 and 0.032, respectively, given by Student’s *t*-test).

**Table 1 viruses-17-00862-t001:** Detailed Ct values obtained by RT-PCR of the swab samples collected in this study.

ID Horse	Day 1		Day 2		Day 3		Day 4		Day 5		Day 6		Day 7		Day 8	
	NS	OS	NS	OS	NS	OS	NS	OS	NS	OS	NS	OS	NS	OS	NS	OS
1	NC	NC	NC	NC	-	-	36.5	-	-	-	33.8	-	35.8	-	-	-
2	NC	NC	NC	NC	-	-	34.4	-	36.7	-	38.3	-	-	-	-	-
3	31.3	34.5	32.2	36.1	NC	-	32.9	-	29.1	-	33.3	-	30.2	34.7	30.9	36.8
4	33.5	35.5	32.8	39.9	38.5	-	33.4	35.2	35.8	37.9	31.4	29.9	37.6	33	29.4	33.8
5	28.0	34.3	19.5	32.6	23.5	25.9	32.6	-	31.0	-	20.0	NC	37.9	35	27.3	-
6	29.1	-	NC	37.2	NC	32	31.3	NC	21.9	29.2	30.0	21.3	34.8	36.7	32.6	33.3
7	32.3	34.9	34.3	36.9	35.6	36	NC	NC	31.4	38.4	-	-	-	-	37.1	NC
8	39.2	38.0	32.6	35.9	27.5	37.4	39.8	-	-	31.7	36.8	36.5	35.8	-	-	NC
9	-	-	-	-	37.1	NC	31.3	24.8	-	-	38	NC	34.5	NC	34.7	NC
10	29.7	38.1	31.8	37.1	-	37.5	30.7	-	36.4	NC	33.5	NC	-	NC	NC	NC

NS: Nasal swab; OS: Ocular swab; NC: Non-collected; -: Negative.

**Table 2 viruses-17-00862-t002:** Detailed RT-PCR results for the detection of EHV-1 in the nasal and ocular swab samples. Positivity ratio (%) and Ct range (mean) are shown.

Day After the Onset of Disease (Stage of Infection)	qPCR Results Nasal Swab Samples		qPCR Results Ocular Swab Samples	
	Pos/Total (%)	Ct Range (Mean)	Pos/Total (%)	Ct Range (Mean)
1–8 days (Whole surveillance period)	56/69 (81.16%)	19.5–39.8 (32.63)	34/63 (53.97%)	21.3–39.9 (34.65)
1–4 days (Early stage of infection)	27/33 (81.81%)	19.5–39.8 (32.31)	20/33 (60.60%)	24.8–39.9 (35.03)
5–8 days (Advanced stage of infection)	29/36 (80.56%)	20.0–38.3 (33.01)	14/30 (46.66%)	21.3–38.4 (33.42)

## Data Availability

The data that support the findings of this study are available on request from the corresponding author. The data are not publicly available due to privacy or ethical restrictions.

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
