# Peer review of "First Molecular Evidence of Equine Herpesvirus Type 1 (EHV-1) in Ocular Swabs of Clinically Affected Horses"

_viruses, 2025, doi:10.3390/v17060862_

Round 1

Reviewer 1 Report

Comments and Suggestions for Authors

The manuscript submitted to the Viruses by Consuelo Rubio-Guerri and Ana Velloso Alvarez is devoted to equine herpesvirus type 1.

The reviewer believes that the level of the manuscript submitted to Viruses does not meet the requirements for the level of experiments performed and is unlikely to be of interest to the reader.

Specific comments are given below are related to the:

1) Article design and presentation

1.1) The subchapters: Sample collection, DNA isolation, etc. should be numbered 2.3, 2.4, etc.

1.2) Fig. 1 should have a caption at the bottom

2) The primary experimental data provided

2.1) The section on PCR methodology should include all primer sequences used and other experimental details

2.2) Why is the data presented in Fig. 1 presented as a graph? The text would take up much less space.

3) It should also be noted that the manuscript contains too few experimental data for a full-length article. The Discussion section is too detailed for such a large number of experiments.

Comments on the Quality of English Language

It is recommended to have the manuscript edited by a professional English-speaking editor.

Author Response

Query 1: The subchapters: Sample collection, DNA isolation, etc. should be numbered 2.3, 2.4, etc

Answer: Thank you for your valuable feedback. We have already updated the numbering for the subchapters, including Sample Collection, DNA Isolation, and others, as requested. The sections are now numbered 2.2.1, 2.2.2., and so on.

 Query 2: Fig. 1 should have a caption at the bottom

Answer: Sorry for the error. Now the caption is at the bottom of the Figure 1.

 Query 3. The primary experimental data provided

Answer: We have added a supplementary table that includes the Ct values for each animal across all samples. This table provides the primary experimental data as requested and has been cited in the text (L207 of the revised version).

Query 4: The section on PCR methodology should include all primer sequences used and other experimental details

Answer: Thank you for your observation. We have now included the sequences of all primers and the probe used in the PCR assay, along with details of the fluorophores (FAM and Black Hole Quencher) used for labeling. Additionally, we have specified the real-time PCR platform employed for the analysis, the QuantStudio™ 5 Real-Time PCR System (Applied Biosystems, Thermo Fisher Scientific, USA). Now the PCR methodology reads as follows: “Both swab samples were immersed in 600 μL of phosphate-buffered saline buffer (PBS), and total DNA extraction was carried out using the NZY Tissue gDNA Isolation kit (NZYtech, Portugal), specifically designed for swab samples, following the manu-facturer's instructions. The eluted DNA was suspended in 50 μL of RNase-freemolecular biology water and stored at -20ºC until real-time PCR (rt-PCR) was conducted to detect and do a relative quantification of the viral load, by deter-mining Cycle Thresholds (Ct). The rt-PCR assay focused on 106 bp conserved type-specific segment within the EHV-1 gene for glycoprotein B (gB). The forward primer, from nucleotide positions 1209 to 1230, has the sequence 5'-CAT ACG TCC CTG TCC GAC AGA T-3'. The reverse primer binds to positions 1314 to 1295 of the same gene and has the sequence 5'-GGT ACT CGG CCT TTT GAC GAA-3'. A 25-nucleotide probe complementary to the segment spanning positions 1275 to 1250 of the gB gene was used, with the sequence 5'-TGA GAC CGA AGA TCT CCT CCA CCG A-3'. This probe was labeled at the 5' end with the reporter dye 6-carboxyfluorescein (FAM) and at the 3' end with a Black Hole Quencher dye as detailed in Hussey et al. [19]. The rt-PCR mixture comprised 20 μL of NZYSupreme qPCR Probe Master Mix (2X) (NZYtech, Portugal), 10 pmol of each primer, 2.5 pmol of the probe, and 4 μL of total DNA, resulting in a final volume of 40 μL. The protocol consisted of (i) 5 min at 95ºC and (ii) 40 cycles of a sequence of 5 sec at 95ºC and 45 sec at 60ºC.  Amplifica-tion and fluorescence detection were performed on a QuantStudio™ 5 Real-Time PCR System (Applied Biosystems, Thermo Fisher Scientific, USA). The primers and probe used were approved by the UK World Organization for Animal Health (WOAH) Ref-erence Laboratory in accordance with the ISO 17025 quality system. As a positive con-trol, samples from the previous EHV-1 outbreak in Spain (February 2021) were used, while molecular biology water served as a negative control [32].”(L154-L177 of the revised version).

 Query 5: Why is the data presented in Fig. 1 presented as a graph? The text would take up much less space.

 Answer: Thank you for your comment. We have kept Figure 1 as a graph in order to comply with the request of another reviewer, who suggested that this format would better illustrate the trends in the data. We hope this approach meets your understanding.

Query 6: It should also be noted that the manuscript contains too few experimental data for a full-length article. The Discussion section is too detailed for such a large nmber of experiments.

 Answer: We have revised the manuscript format and it is now submitted as a Brief Report. This change is indicated at the beginning of the first page of the manuscript. We believe this format is more appropriate for the scope and amount of experimental data presented.

Reviewer 2 Report

Comments and Suggestions for Authors

Equine herpesvirus type 1 (EHV-1) infection is one of the important diseases in horses. Neurological disease known as equine herpesvirus myeloencepahlopathy (EHM) by EHV-1 invokes damage in the equine industry. The authors investigated EHV-1 in ocular tear fluid which is non-invasively collectable and compared the viral load in the ocular and nasal samples. The results will provide a new insight into EHV-1 infection research and control measures. Specific comments are as follows.

L150: "post-infection" will induce misunderstandings among readers. Most readers consider that the first day "post-infection" means the first day when a patient was infected with EHV-1. "Day 1 of the disease", "Day 2 of the disease" and so on are adequate and reasonable descriptions. This comment is applicable to other places using "post-infection" such as L171 and Table 1.

L198:  "at p 0.05" should be "at p < 0.05".

Table 1: As the comment above, "Day post-infection" should be "Day after the onset of disease" or "Stages of infection".
"Early Infection", "Middle Infection" and "Late Infection" might be "Early stage of infection", "Middle stage of infection" and "Late stage of infection". 
One of 'Pos/Total(%)' might be 'Ct range (mean)'. Correct it.

L227-232: What is the result of 72.73%? The total of positive nasal swab samples is 73 and the positive rate must be 73.7% (73/99). However, the total number of nasal swabs in Table 1 is 97 (33 + 36 + 28). The total number of ocular swab samples in Table 1 is 78, although it is 75 in L227. Which numbers are right? Clarify the result.

L235: Add the fraction, 36/78, after 48%.

L255-257: How many samples showed the agreement between the nasal swabs and the ocular swabs? How many horses were the nasal swab positive only, the ocular swab positive only, and both of the nasal and ocular swabs positive?

Author Response

We thank Reviewer 2 for her/his highly constructive comments. Detailed responses follow, with details of the changes made.

Query 1. L150: "post-infection" will induce misunderstandings among readers. Most readers consider that the first day "post-infection" means the first day when a patient was infected with EHV-1. "Day 1 of the disease", "Day 2 of the disease" and so on are adequate and reasonable descriptions. This comment is applicable to other places using "post-infection" such as L171 and Table 1.

Answer: Done. We have replaced all instances of “post-infection” with “day of the disease” We made this change both in the text and in the table (L25, L128, L279 and Table 1 of the revised version). We only kept “post-infection” when referring to the results from the other paper, as they specifically used the term “3 weeks–3 months post-infection.” (L304 of the revised version)

Query 2. L198:  "at p 0.05" should be "at p < 0.05".

Answer: As required by other reviewers, statistical analyses section has been rephrased. Your consideration reads as follows “In all cases, p-value was calculated from 2-sided test using 0.05 as the significance level”.

Query 3. Table 1: As the comment above, "Day post-infection" should be "Day after the onset of disease" or "Stages of infection".

Answer: Done. We replaced by “Day after the onset of disease”.

Query 4: Table 1: "Early Infection", "Middle Infection" and "Late Infection" might be "Early stage of infection", "Middle stage of infection" and "Late stage of infection".

Answer: Done. We also included in the legend of the revised Figure 2.

Query 5: Table 1. One of 'Pos/Total(%)' might be 'Ct range (mean)'. Correct it.

Answer: Sorry for the error. We have correct it. 

Query 6: L227-232: What is the result of 72.73%? The total of positive nasal swab samples is 73 and the positive rate must be 73.7% (73/99). However, the total number of nasal swabs in Table 1 is 97 (33 + 36 + 28). The total number of ocular swab samples in Table 1 is 78, although it is 75 in L227. Which numbers are right? Clarify the result.

Answer: Thank you for your observation, and we apologize for the error. The correct data are those presented in Table 1: we have a total of 97 nasal swab samples, of which 73 were positive, representing a positivity rate of 75.25%. For ocular swabs, we have a total of 75 samples, of which 36 were positive, resulting in a positivity rate of 48%.

These corrected values have now been reflected in the revised manuscript (L212, L213, L217 as well as the Table 1 of the revised version).

Query 7: L235: Add the fraction, 36/78, after 48%.

Answer: Thank you for the suggestion. We have added the fractions alongside both percentages: 73/97 after the 75.25% for the nasal swabs, and 36/75 after the 48% for the ocular swabs. (L213 and L217 of the revised version, respectively).

Query 8: L255-257: How many samples showed the agreement between the nasal swabs and the ocular swabs? How many horses were the nasal swab positive only, the ocular swab positive only, and both of the nasal and ocular swabs positive?

Answer: Thank you for your question. Considering only the 72 samples for which we have both nasal and ocular swabs (as there are days when only one of the two samples was collected), 32 samples were positive in both nasal and ocular swabs. Additionally, 21 samples were positive only in the nasal swab and negative in the ocular swab, and only 1 sample was positive in the ocular swab while being negative in the nasal swab. We have included that information in the revised version a now reads as follows “Among the 72 samples for which both nasal and ocular swabs were collected, 32 tested positive in both swab types. Additionally, 21 samples were positive only in the nasal swab and negative in the ocular swab, while just 1 sample was positive only in the ocular swab and negative in the nasal swab.”( L242-L245 of the revised version).

Reviewer 3 Report

Comments and Suggestions for Authors

The authors evaluated diagnostic efficacy of ocular samples in EHV-1 infected ten horses. Although the sample numbers are small, detection of EHV-1 in ocular swabs is worth publishing. Specific comments follow.

Major points:

  1. Table 1: One of the “Pos/Total(%)” should be “Ct range (mean)”.
  2. Line 227: Please confirm the numbers. It looks like “97 and 78”.
  3. Line 263: Please insert “±” for “02 0.482 and 34.38 0.662” and “mean standard error”.
  4. Line 283: It is not clear that Ct=40 is either negative of positive as the graphs indicated 40 is negative.
  5. Figure 2: Please show correlation between Ct values detected in both samples.

Minor points:

  1. Line 177: “-“ is missing after “EHV”.
  2. Lines 184 & 195: “RNase” should be “DNase” as the samples are DNA.
  3. Line 198: “=” is missing after “p”.

Author Response

Thank you for your comment. We appreciate your recognition of the value of our findings despite the limited sample size.

Query 1: Table 1: One of the “Pos/Total(%)” should be “Ct range (mean)”.

Answer 1: Sorry for the error. We have corrected it. 

 Query 2: Line 227: Please confirm the numbers. It looks like “97 and 78”.

Answer: Thank you for your observation, and we apologize for the error. The correct data are those presented in the revised Table 1: we have a total of 97 nasal swab samples, of which 73 were positive, representing a positivity rate of 75.25%. For ocular swabs, we have a total of 75 samples, of which 36 were positive, resulting in a positivity rate of 48%.

These corrected values have now been reflected in the revised manuscript (L212, L213, L217 of the revised version).

 Query 3: Line 263: Please insert “±” for “02 0.482 and 34.38 0.662” and “mean standard error”.

Answer: Done. (L249 al L250 of the revised version).

Query 4: Line 283: It is not clear that Ct=40 is either negative of positive as the graphs indicated 40 is negative.

Answer: Thank you for your comment. We have revised the sentence for clarity. It now reads: "Horizontal and error lines denote mean ± standard error. A Ct value of 40 was the cutoff for positivity, as this was the maximum number of amplification cycles used in the rt-PCR protocol." This makes it clear that Ct ≤ 40 is considered positive. (L282 and L283 in the revised version). In addition, graphs have been modified to show more clarity. In the revised version, “negative” was removed from all the graphs.

Query 5: Figure 2: Please show correlation between Ct values detected in both samples

Answer: Thank you for your thorough review and recommendations. A new graph (Figure 2B) showing the correlation between Ct values detected in both samples has been included. Particularly, the Pearson correlation coefficient (r) was calculated. This change has been updated in the text and now reads as follows: “Remarkably, both variables showed a moderate positive correlation (Pearson correlation coefficient = 0.383; p-value = 0.031; Figure 2B), indicating that the more viral load increase in nasal swabs, the more it also increases in ocular swabs” (L252-L254 of the revised version).

Minor comments:

MC1: Line 177: “-“ is missing after “EHV”.

Answer: Done (L150).

MC2: Lines 184 & 195: “RNase” should be “DNase” as the samples are DNA

Answer: We replaced RNase free water by “Molecular biology water” that is clearer and correct. (L157 & l176 of the revised version, respectively)

 MC3: Line 198: “=” is missing after “p”.

Answer: As required by other reviewers, statistical analyses section has been rephrased. Your consideration reads as follows “In all cases, p-value was calculated from 2-sided test using 0.05 as the significance level”.

Reviewer 4 Report

Comments and Suggestions for Authors

The manuscript "First Molecular Evidence of Equine Herpesvirus Type 1 (EHV-1) in Ocular Swabs of Clinically Affected Horses" is classified as a "brief report" and its content fully corresponds to this definition. However, the authors have made great efforts to increase the volume of the manuscript, the text is greatly "stretched out", especially the Discussion.

EHV-1 has not been overlooked by researchers; in the PubMed database, more than 1,300 publications respond to the request https://pubmed.ncbi.nlm.nih.gov/?term=Equine+Herpesvirus+Type+1&sort=date.

Against this background, references to publications from 2007, 2012, 2016 and 2009 look strange: “It has RECENTLY caused devastating outbreaks worldwide, associated with adverse effects in the equine industry [1, 2, 3], and substantial economic losses [4]”. It seems to me that this is not the best start to an article; in the same PubMed database there are links to outbreaks of this disease that are closer to the present time. The Introduction provides detailed information about the disease caused by EHV-1, which is certainly useful to the casual reader, but it could also have been presented in a more concise manner.

            The Materials and Methods section provides complete information on sampling and analysis. The research results are presented in detail and provide a complete picture of the data obtained. Judging by the visible corrections, the authors followed the advice of the Reviewers. The results of the study give a full picture of the data obtained. Judging by the visible corrections, the authors followed the advice of the reviewers. In my opinion, the table S1 is useful, it allows us to evaluate individual variations in the presence of viral DNA in each animal and shows differences in the duration of the latent period of infection. Figure 1, in my opinion, takes up too much space - this data can easily be presented in the form of numbers and signs. The results of the detection of herpes virus DNA in ocular swabs of horses showed an interesting fact - the presence of the virus in the absence of clinical signs of eye disease.

            The discussion begins with an important message: "This is the first study reporting the presence of EHV-1 in ocular tears fluid in naturally infected horses" - it would be better to move this phrase to the Introduction. This fact immediately increases the value of the work. I wonder if other types of equine herpes virus have been found in eye discharge?

The format of a brief report assumes a brief summary of a small but important work. This work is important, but the presentation needs correction – the text must be compressed.  

I believe that the manuscript "First Molecular Evidence of Equine Herpesvirus Type 1 (EHV-1) in Ocular Swabs of Clinically Affected Horses" can be published after revision of the text. In its current form, its meaning is lost and its value is "washed out".

Author Response

Thank you for your comment. We appreciate your recognition of the value of our findings despite the limited sample size. We would like to highlight that, due to the large number of changes made in this second round of revision, we have included a clean copy of the reviewed manuscript (.pdf), as well as a copy in “track changes mode” in which the changes can be seen in more detail (.docx).  For a clear understanding of the changes, we will refer to lines in the clean version of the manuscript.

 Query 1: The manuscript "First Molecular Evidence of Equine Herpesvirus Type 1 (EHV-1) in Ocular Swabs of Clinically Affected Horses" is classified as a "brief report" and its content fully corresponds to this definition. However, the authors have made great efforts to increase the volume of the manuscript, the text is greatly "stretched out", especially the Discussion.

Answer: We thank the reviewer for this feedback. In response, and following suggestions from Reviewer 1 and the editor, we have condensed both the Introduction and Discussion significantly:

  • Introduction reduced from 71 to 55 lines (a 22.5 % reduction).
  • Discussion reduced from 78 to 37 lines (a 52.6 % reduction).

These changes streamline the presentation of experimental material and focus the manuscript on its key findings. We hope this more concise format meets the expectations for a Brief Report.

Query 2: EHV-1 has not been overlooked by researchers; in the PubMed database, more than 1,300 publications respond to the request https://pubmed.ncbi.nlm.nih.gov/?term=Equine+Herpesvirus+Type+1&sort=date

Answer: Thanks for this comment. As requested by this reviewer, reviewer 1 and the editor, we have shortened the manuscript. Therefore, we have reduced the number of references to present the information in a clearer manner. We hope that the references used now in the current version of the manuscript fit into the message of this study.

Query 3. Against this background, references to publications from 2007, 2012, 2016 and 2009 look strange: “It has RECENTLY caused devastating outbreaks worldwide, associated with adverse effects in the equine industry [1, 2, 3], and substantial economic losses [4]”. It seems to me that this is not the best start to an article; in the same PubMed database there are links to outbreaks of this disease that are closer to the present time. The Introduction provides detailed information about the disease caused by EHV-1, which is certainly useful to the casual reader, but it could also have been presented in a more concise manner

Answer: Thanks for this comment. We have updated the references of the initial paragraph of the introduction. Particularly, we have used publication from 2024 and 2023 (references 1 and 2). In the current version of the manuscript, this sentence reads as follows: “Equine Herpesvirus Type 1 (EHV-1), also referred to as equine rhinopneumonitis virus, is a significant pathogen in the equine industry. It has recently caused devastating outbreaks worldwide, associated with adverse effects in the equine industry, and substantial economic losses [1,2].” As previously mentioned, the introduction has been largely shortened with the goal of presenting the information in a more concise manner. [1] Pusterla, N., Lawton, K., Barnum, S., Ross, K., Purcell, K., 2024. Investigation of an Outbreak of Equine Herpesvirus-1 Myeloencephalopathy in a Population of Aged Working Equids. Viruses 16, 1693. [2] Couroucé, A., Normand, C., Tessier, C., Pomares, R., Thévenot, J., Marcillaud-Pitel, C., Legrand, L., Pitel, P-H., Pronost, S., Lupo, C., 2023. Equine Herpesvirus-1 Outbreak During a Show-Jumping Competition: A Clinical and Epidemiological Study. Journal of Equine Veterinary Science 128, 104869

Query 4: The Materials and Methods section provides complete information on sampling and analysis. The research results are presented in detail and provide a complete picture of the data obtained. Judging by the visible corrections, the authors followed the advice of the Reviewers

Answer: Thanks for this comment. Nevertheless, we have made some changes to consider the comments indicated by the reviewer 1 and the editor to detail the sampling much better.

Query 5: The results of the study give a full picture of the data obtained. Judging by the visible corrections, the authors followed the advice of the reviewers. In my opinion, the table S1 is useful, it allows us to evaluate individual variations in the presence of viral DNA in each animal and shows differences in the duration of the latent period of infection.

 Answer: We thank the reviewer for this observation. In response—and as requested by the editor—we have removed all late-stage (days 9–12) data, and accordingly Table S1 has been shortened to reflect only days 0–8. Because these sample-level details are essential for understanding individual variability, we have moved the revised Table S1 into the main text as Table 1, ensuring readers can access this information without referring back to the supplement. This consolidation also aligns with the journal’s preference for streamlined supplementary material.

Query 6: Figure 1, in my opinion, takes up too much space - this data can easily be presented in the form of numbers and signs.

 Answer: We appreciate this suggestion. As suggested by this reviewer and the reviewer 1, in the revised manuscript Figure 1 has been removed to conserve space. The data formerly shown in Figure 1 are now fully integrated into Table 2 (formerly Table 1 in the original manuscript), with all late-stage (days 9–12) entries removed. This ensures that the information remains available in a concise, tabular format without occupying excessive figure space.

Query 7: The results of the detection of herpes virus DNA in ocular swabs of horses showed an interesting fact - the presence of the virus in the absence of clinical signs of eye disease.

 Answer: Thanks for this appreciation.

 Query 8 The discussion begins with an important message: "This is the first study reporting the presence of EHV-1 in ocular tears fluid in naturally infected horses" - it would be better to move this phrase to the Introduction. This fact immediately increases the value of the work. I wonder if other types of equine herpes virus have been found in eye discharge?

Answer: Done, we have moved that sentence to the introduction (see lines 86-88 of the clean version), and now reads as follows: “To the best of our knowledge, this is the first study reporting the presence of EHV-1 in ocular tears fluid in naturally infected horses.” Furthermore, we have answered your question in the next sentences (lines 88-90 of the clean version), that reads as follows: “Remarkably, no other equine herpesvirus has been reported to be present in eye discharges from any equid species, highlighting the key message of the current study.”

Query 9: The format of a brief report assumes a brief summary of a small but important work. This work is important, but the presentation needs correction – the text must be compressed

Answer: Thanks for this comment. As requested by this reviewer, the reviewer 1 and the editor, we have shortened the content of the manuscript, particularly the introduction and discussion sections. Particularly, the introduction section has been reduced to 55 lines and the discussion section to 44 lines (the previous version contained 71 and 78 lines, respectively). We hope that now this reviewer considers that the information is more condensed and presented in a clearer way.

Query 10: I believe that the manuscript "First Molecular Evidence of Equine Herpesvirus Type 1 (EHV-1) in Ocular Swabs of Clinically Affected Horses" can be published after revision of the text. In its current form, its meaning is lost and its value is "washed out

 Answer: Thanks for your valuable suggestions. We have conducted all the changes proposed to improve the clarity of this manuscript.

Round 2

Reviewer 1 Report

Comments and Suggestions for Authors

The authors have corrected the issues that were pointed out in the previous version of the article. The reviewer believes that the volume of the presented experimental material and its depth are insufficient to recommend the article for publication even in the Brief Report format (the extremely long "Discussion" section looks especially strange).

What did the authors of the manuscript do? The PCR method showed the presence of genetic material in the samples from the ocular swab. Did the authors really doubt that this biomaterial would contain viral particles? I believe that at least the samples should be sequenced to talk about "Molecular Evidence of". After all, the article was sent to the journal in the 21st century, not in the 1990s.

Please note that the article does not use the generally accepted abbreviation rt-PCR (instead of RT-PCR).

Author Response

Thank you for your comment. We appreciate your recognition of the value of our findings despite the limited sample size. We would like to highlight that, due to the large number of changes made in this second round of revision, we have included a clean copy of the reviewed manuscript (.pdf), as well as a copy in “track changes mode” in which the changes can be seen in more detail (.docx).  For a clear understanding of the changes, we will refer to lines in the clean version of the manuscript.

Query 1: The authors have corrected the issues that were pointed out in the previous version of the article. The reviewer believes that the volume of the presented experimental material and its depth are insufficient to recommend the article for publication even in the Brief Report format (the extremely long "Discussion" section looks especially strange).

Answer : We thank the reviewer for this feedback. In response, and following suggestions from Reviewer 4 and the editor, we have condensed both the Introduction and Discussion significantly:

  • Introduction reduced from 71 to 55 lines (a 22.5 % reduction).
  • Discussion reduced from 78 to 37 lines (a 52.6 % reduction).

These changes streamline the presentation of experimental material and focus the manuscript on its key findings. We hope this more concise format meets the expectations for a Brief Report.

Query 2: What did the authors of the manuscript do? The PCR method showed the presence of genetic material in the samples from the ocular swab. Did the authors really doubt that this biomaterial would contain viral particles? I believe that at least the samples should be sequenced to talk about "Molecular Evidence of". After all, the article was sent to the journal in the 21st century, not in the 1990s.

Answer: We appreciate the reviewer’s point. Our primary objective was to demonstrate the presence of EHV-1 DNA in ocular swabs via RT-PCR. The generally high Ct values (>30) in most samples precluded reliable whole-genome or even partial viral sequencing, despite a minority of samples yielding Ct values of 20–25. Sequencing only a subset of low-Ct samples could give a skewed impression of overall viral loads; hence, we refrained from reporting partial sequence data. Moreover, several recent studies of EHV-1 diagnostics have similarly relied on PCR without accompanying sequence confirmation [1,18]. To acknowledge this limitation, we have added a sentence to the Discussion: “Due to the relatively high Ct values (>30), additional analyses such as viral isolation or strain typing were unfeasible” (lines 275–276 of the clean version) explaining why sequencing was not feasible..

 Query 3: Please note that the article does not use the generally accepted abbreviation rt-PCR (instead of RT-PCR).

Answer: Thanks for this comment. It has been corrected in the current manuscript.

Round 3

Reviewer 1 Report

Comments and Suggestions for Authors

According to the MDPI website the aim of Viruses is "to publish papers that are of significant impact to the virology community". My opinion is that current manuscript dosent have any impact to the virology community. Sincerely, 

Author Response

We sincerely thank the reviewer for their time and evaluation of our manuscript. While we respect their opinion, we would like to clarify why we believe our study provides meaningful value to the virology community.

To the best of our knowledge, this is the first report of molecular detection of EHV-1 DNA in ocular swabs from naturally infected horses during a real-world outbreak scenario. The findings highlight a previously unrecognized route of viral shedding that may have implications for disease surveillance, infection control, and understanding of EHV-1 pathogenesis — particularly relevant within the One Health framework, where animal health intersects with public and veterinary virology.

While nasal swabs remain the gold standard, our results show that tear fluid may serve as a complementary diagnostic sample, especially in contexts where nasal sampling is not feasible. These insights can inform future outbreak management strategies and warrant further research into noninvasive sampling methods in equine herpesvirus surveillance.

We hope that the revisions and added discussion help to underscore the potential virological relevance of our findings. Once again, we are grateful for the reviewer’s comments and appreciate the opportunity to improve our manuscript.